# Lottery Ticket Structured Node Pruning for Tabular Datasets

## Abstract

In this paper we presented two pruning approaches on tabular neural networks based on the lottery ticket hypothesis that went beyond masking nodes by resizing the models accordingly. We showed top performing models in 6 of 8 datasets tested in terms of F1/RMSE. We also showed in 6 of 8 datasets a total reduction of over 85% of nodes and many over 98% reduced with minimal affect to accuracy. In one dataset the model reached a total size of one node per layer while still improving RMSE compared to the larger model used for pruning. We presented results for two approaches, iterative pruning using two styles, and oneshot pruning. Iterative pruning gradually reduces nodes in each layers based on norm pruning until we reach the smallest state, while oneshot will prune the model directly to the smallest state. We showed that the iterative approach will obtain the best result more consistently than oneshot.

## 1 Introduction

It has been understood for a while that pruning a network after training can lead to much smaller networks that still maintains similar accuracy to the original network. However the question arises, if these smaller networks perform just as well as the larger ones, could one not train the smaller network architecture to begin with?

The lottery ticket hypothesis states, "a randomly-initialized, dense neural network contains a subnetwork that is initialized such that - when trained in isolation - it can match the test accuracy of the original network after training for at most the same number of iterations" Frankle & Carbin (2019). These smaller networks are referred to as "winning tickets". The process of finding winning tickets was proposed to start with an initial set of weights, train the network, prune the network, and finally reset the remaining weights back to their initial state.

It has been shown that these winning tickets perform as well, or sometimes better and require less time to train when compared to the original network. In this paper we focus on applying this hypothesis to tabular neural networks for several datasets to generate smaller networks with similar performance.

We use tabular neural networks from FastAI Howard et al. (2018) which work on the principle of converting categorical features into embeddings. The models come with strong features such as normalization of continuous features, filling in missing data, and converting dates into categorical features. The models can be trained with their cycle training approach which raises and lowers the learning rate to find the best local minimum while training.

There are many small tabular datasets which often don't get optimized in research. These datasets are filled with categorical features and missing entries while often having few samples to test. In this paper we use the lottery ticket hypothesis to improve FastAI's tabular neural network to easily prune models for any tabular dataset. We test a wide range of datasets from large to small including different ranges of categorical and continuous features.

This paper is organized as follows. Section 2 describes related previous works. Section 3 presents the datasets used and our pruning methodology. Experiments, results and comparisons are presented in Section 4. Finally, the paper is concluded in Section 5.

## 2 Previous works

The lottery ticket hypothesis was first tested by Frankle & Carbin (2019) on dense neural networks and convolutional neural networks for MNIST and CIFAR10. The authors achieved networks 80-90% smaller while both creating faster networks and increasing the accuracy of the models in comparison to the original sized models. The authors show that while pruning has been around for a long time, pruning networks often lead to more difficult training with less accuracy. By finding the lottery tickets of a network, difficulty in training of smaller networks is reduced while accuracy can be maintained and even exceed the original model.

Since its introduction, there have been many papers testing the lottery ticket hypothesis on different architectures. Morcos et al. (2019) look to find winning tickets in networks by searching for common winning tickets across many datasets. They test on a range of image datasets such as Fashion MNIST, SVHN, CIFAR-10/100, ImageNet, and Places365. Their findings were that winning tickets generalized over many datasets, but interestingly the larger datasets produced better winning tickets that were more generalizable than those from smaller datasets. Girish et al. (2021) use the lottery ticket hypotheses for object detection achieving a sparsity of up to 80%.

Chen et al. (2021) test the lottery ticket hypothesis on ever growing graph neural networks (GNN). For node classification tasks, they show a decrease in multiply-accumulate operations (MACs) of up to 98% while having a sparsity of nearly 98% with minimal affect to performance. For linking predictions, they show similar sparsity with a large reduction in MACs.

The field of natural language processing (NLP) has recently exploded with new models since the introduction of the transformer architecture Vaswani et al. (2017). The models grow increasingly large, from sizes like the popular BERT Devlin et al. (2018) architecture of up to 350 million parameters to sizes like GPT-3 Brown et al. (2020) reaching up to 175 billion parameters. Chen et al. (2020) prune the BERT architecture, but they begin from the pretrained state and test the lottery ticket hypothesis using downstream tasks. The paper only applies unstructured pruning stating "since we perform a scientific study of the lottery ticket hypothesis rather than an applied effort to gain speedups on a specific platform, we use general-purpose unstructured pruning" Chen et al. (2020). With unstructured pruning, they are able to find networks with sparsity of 40-90% while testing a range of downstream tasks.

Tabular models are heading towards larger networks as well with the recent transformer architecture making its way to tabular data. Padhi et al. (2021) proposed a time series BERT architecture for tabular data called TabBERT and a tabular data generator TabGPT based on the GPT architecture. Gu & Budhkar (2021) use transformers to convert text in tabular data into features and released a package to process the data. TabTransformer by Huang et al. (2020) uses the transformer architecture to achieve new SOTAs and tested on 15 datasets, competing well with ensemble approaches.

In this paper we focus on tabular dense neural networks with the goal to prune them as much as possible while maintaining accuracy utilizing the lottery ticket hypothesis. We test our models on 8 datasets of different varieties and show competing results for datasets with available comparisons.

## 3 Datasets

In this section we describe the tabular datasets used in our experiments. We selected a wide range of datasets from as few hundred samples to millions. Each dataset has a variety of tabular features, some containing just continuous features, others containing only categorical features, and many mixed with both. The datasets are described in more detail in this section and we present the number of samples, number of features, evaluation metrics, and train-validation-test splits used in our experiments in Table 1.

The first dataset is the Alcohol dataset Cortez & Silva (2008). It is our smallest dataset and contains the most features with a mix of continuous and categorical features. The dataset contains information on students such as school, gender, age, information like hobbies and goals, their family and related information such as work, education, size, etc. As quoted from this recent dataset survey Mihaescu & Popescu (2021), "This dataset has also been uploaded on Kaggle where 305 publicly available kernels perform exploratory data analysis. Unfortunately, there is not defined any task with specific validation metric such that there is no leaderboard publicly available", so we used

Table 1: Dataset Information. We provide the dataset names (Video Games abbreviated to Games and Wine Quality abbreviated to Wine), the evaluation metric used, the number of features (categorical and continuous), the number of samples and the percentage of train/valid/test split.

| Dataset | | | # Samples | | | % Split | | |
|---|---|---|---|---|---|---|---|---|
| Name | Metric | Cat \| Cont | Train | Valid | Test | Train | Valid | Test |
| Alcohol | RMSE | 17 \| 10 | 253 | 63 | 79 | .64 | .16 | .20 |
| Games | RMSE | 4 \| 4 | 10623 | 2655 | 3320 | .64 | .16 | .20 |
| Wine | RMSE | 0 \| 11 | 1024 | 255 | 320 | .64 | .16 | .20 |
| Chocolate | RMSE | 6 \| 1 | 1149 | 287 | 359 | .64 | .16 | .20 |
| Poker | RMSE | 10 \| 0 | 20008 | 5002 | 1000000 | .02 | .005 | .975 |
| Titanic | F1 | 6 \| 2 | 463 | 154 | 155 | .60 | .20 | .20 |
| Health | F1 | 5 \| 5 | 59724 | 14932 | 18664 | .64 | .16 | .20 |
| Susy | F1 | 0 \| 18 | 4000000 | 500000 | 500000 | .80 | .10 | .10 |

the workday alcohol consumption of the student as the final goal. The model aims to predict the students' workday alcohol consumption which is a target range of 1 to 5 where 1 is very low and 5 is very high consumption.

The next dataset is the Video Games Sales dataset[1]. The dataset also doesn't have a clear final goal or leaderboard information. There are features of videos games such as rank, publisher, year and Genre and sales information. We have information on sales for North America, Europe, Japan, other countries and global sales. Predicting global sales means we would have to omit information about sales in other countries as it would be just a simple sum of those sales, so we decided to predict North American sales given information on the video game and sales information in other countries not including global sales.

The Wine Quality dataset Cortez et al. (2009) aims to predict a quality score between 0 and 10 of the wine given its features. The features are continuous values representing different acidity rates, sugar levels, density, pH and more.

The Chocolate Ratings dataset[2] was created to generate expert opinions on chocolate. We must predict the expert ratings which are values between 1 and 5 where 1 is bad taste and 5 is the best taste. The features include the company, location, type of beans, percentage of cocoa, and origin information.

The Poker Hand dataset Cattral et al. (2002) is a very large set of poker hands. Each sample is a set of 5 cards indicating the card numbers as 5 features and their suits as 5 more features. The final goal of this dataset is to predict the poker hand such as 0 for nothing, 1 for one pair, 2 for two pairs, 3 for three of a kind, 4 for a straight, 5 for a flush, 6 for a full house, 7 for four of a kind, 8 for straight flush, and 9 for a royal flush. We used this as an regression problem where higher (9) the better hand and lower (0) the worse the hand.

The Titanic dataset[3] uses information on passengers of the Titanic to predict whether they survived. The features include gender, cabin, location of embarkment, ticket class (1st, 2nd, 3rd), number of siblings or spouses, number of parents or children, age and fare. The goal is to predict the survival of the individual (yes or no) using F1 as a metric. There is a predefined test set without labels which must be submitted through kaggle to be evaluated, but our results reflect a train/validation/test split from the labelled train set only. We also take our best available model for this dataset and run it through kaggle to get a test score for their test set in the experiments section. Note that a new dataset Titanic Extended[4] was introduced with many more features and much less empty features derived from the literature allowing others to achieve 100% accuracy, so we opted to use the more difficult prior version for testing without knowledge of the extended features.

---

[1]https://github.com/GregorUT/vgchartzScrape, https://www.kaggle.com/gregorut/videogamesales
[2]https://www.kaggle.com/rtatman/chocolate-bar-ratings
[3]https://www.kaggle.com/c/titanic/data
[4]https://www.kaggle.com/pavlofesenko/titanic-extended

The Health Insurance dataset[5] aims to predict vehicle insurance sales to customers of health insurance. We are given information on the policy holder such as a unique ID (omitted in training), gender, age, has a driving license, region, types of vehicle information like age and damage, and information on their premiums. The goal is whether the customer will accept the vehicle insurance which is a binary prediction using F1 as a metric.

The Susy dataset Baldi et al. (2014) is our largest dataset containing 4 million training samples, 500k validation and 500k test samples. The test set was predefined for this dataset as the last 500k samples in the list. The dataset contains simulation data of a particle collider with the goal to find rare particles. There are eight kinematic features of the collision and 10 functions of those features with the final goal distinguishing signal from background using F1 as a metric.

## 4  METHODOLOGY

To predict on these datasets, we use tabular neural networks repurposed from FastAI Howard et al. (2018) and using their cycle training procedures. The cycle training increases and reduces the learning rate as needed to reach the best local minimum, so we are able to set default learning rates for all models and automatically train our pruning approaches. We kept the number of training epochs the same for all pruned versions of a dataset but used early stopping to avoid overfitting the models based on validation scores. The data was preprocessed by converting categorical features to embeddings, filling missing values in features and normalizing continuous features. The models were altered to allow pruning of nodes selected by different pruning approaches we describe below. Pytorch provides functionality to select weights to prune with L1 norm pruning, a generic LN norm pruning where we selected N=-inf, and random pruning. Of infinite possibilities for N, we chose the interesting case of N=-inf because while N=1 selects nodes based on the sum of all weights, we didn't want a similar norm such as N=inf which selects the largest weight, so we chose N=-inf which focuses on the smallest weight of each node and prunes based on the largest of these norms (even though technically N=-inf isn't a norm). We pruned all weights associated to nodes from the linear layers of the model depending on a rate P where the pytorch functions provide a mask on the nodes setting all of the weights to 0.

While the pruning functionality provided by pytorch is useful for testing, we wanted the full benefit of smaller models which provide inference and training time improvements rather than a mask over the weights which still require computation. In this case we designed a simple approach to fully prune the weights. First we save the initial untrained weights of the model W0. Then we train the model until we reach the optimal validation stopping point and record the weights W1. Next we generate pruning masks for W1 depending on the selected norm (or random) and pruning rate P, then apply the masks on the initial untrained weights W0. The remaining subset of weights from W0 will be copied over to a smaller blank model sized appropriately for the non-zero weights. Then we continue to train the pruned model depending on the iterative or oneshot styles described below using the smaller network.

FastAI's tabular models start with a concatenation of categorical embeddings and continuous features, then linear layers follow with optional batch normalization. We implemented batch normalization in the pruning process, but results were better without these layers and removing them left us with smaller and faster models.

### 4.1  ITERATIVE PRUNING

A simple iterative approach can be thought of as starting with a model size and reducing it slowly until we reach an optimal size based on criteria for accuracy and speed performance. The iterative approach uses a pruning rate P=0.5 (50%). We describe two iterative pruning styles, one using a large model potentially holding many lottery tickets, and the other using a good performing model as a starting point with potentially better lottery tickets selected (but possibly smaller in size).

The first approach uses a starting point of a large model [1600, 800] where the parameters are the sizes of the linear layers respectively. We call this starting point the original model. We train the

---

[5]https://www.kaggle.com/anmolkumar/health-insurance-cross-sell-prediction

original model, prune it at the rate of P, then repeat on the pruned model until we reach the smallest state possible (ideally [1, 1]).

In the second approach, we instead train at various starting points of different model sizes starting with [1600, 800], then continuing with [800, 400] and so on reducing the size by 50% at each step. We train all of these original models in search of the best starting point for the best performing model. Then we run iterative pruning like in the first approach on this potentially smaller model, but ideally with better lottery ticket weights available to be selected.

## 4.2  ONESHOT PRUNING

The oneshot pruning approach works like iterative, but uses a prune rate of P=1.0. This means we prune the entire model in one shot reducing it down entirely to a size of [1, 1]. We run oneshot on all of the potential starting points we described in iterative (second approach), then we select the best performing oneshot model.

## 5  EXPERIMENTATIONS, RESULTS AND DISCUSSION

There are three components to our results. The first is best performing accuracy, the second is a comparison in train and inference times, and the last is a comparison of model sizes with minimal affect to accuracy. To evaluate the overall performance of our models, we also provide a table of comparisons using several different tabular models. In all tests, all random seeds were set to the same value to allow fair comparisons and reproducibility of any result shown. We paid careful attention to have the model weights set to the same initial states for all styles of pruning.

Table 2: Train times averaged over all epochs. The size column shows the possible prune states representing the number of nodes in each layer of the neural network, each state 50% smaller than the last. Then for each dataset we provide the average epoch train time in seconds followed by the percentage improvement from [1600, 800] in parentheses.

| Size | Alcohol | Games | Wine | Chocol. | Poker | Titanic | Health | Susy |
|---|---|---|---|---|---|---|---|---|
| **1600, 800** | 0.21s (.00) | 35.47s (.00) | 0.90s (.00) | 1.04s (.00) | 19.33s (.00) | 0.38s (.00) | 139.50s (.00) | 3611.18s (.00) |
| **800, 400** | 0.15s (.247) | 12.23s (.655) | 0.49s (.454) | 0.65s (.373) | 12.49s (.354) | 0.25s (.328) | 52.40s (.624) | 1168.57s (.676) |
| **400, 200** | 0.14s (.302) | 6.53s (.816) | 0.39s (.561) | 0.55s (.468) | 10.10s (.477) | 0.23s (.4) | 32.45s (.767) | 529.79s (.853) |
| **200, 100** | 0.14s (.33) | 5.24s (.852) | 0.36s (.595) | 0.54s (.483) | 8.52s (.559) | 0.21s (.437) | 30.28s (.783) | 425.54s (.882) |
| **100, 50** | 0.14s (.322) | 4.57s (.871) | 0.35s (.606) | 0.54s (.478) | 7.77s (.598) | 0.21s (.448) | 27.68s (.802) | 337.41s (.907) |
| **50, 25** | 0.14s (.302) | 4.54s (.872) | 0.34s (.616) | 0.52s (.498) | 7.2s (.627) | 0.21s (.443) | 27.18s (.805) | 314.14s (.913) |
| **25, 13** | 0.14s (.335) | 5.47s (.846) | 0.35s (.615) | 0.51s (.505) | 6.36s (.671) | 0.21s (.439) | 27.37s (.804) | 280.77s (.922) |
| **13, 7** | 0.14s (.324) | 4.30s (.879) | 0.34s (.615) | 0.51s (.507) | 5.07s (.738) | 0.21s (.45) | 27.2s (.805) | 289.77s (.92) |
| **7, 3** | 0.14s (.321) | 4.31s (.879) | 0.34s (.615) | 0.52s (.496) | 4.95s (.744) | 0.21s (.45) | 26.91s (.807) | 287.14s (.92) |
| **3, 1** | 0.14s (.302) | 4.63s (.869) | 0.34s (.622) | 0.52s (.505) | 5.24s (.729) | 0.21s (.449) | 27.2s (.805) | 292.28s (.919) |
| **1, 1** | 0.14s (.339) | 4.47s (.874) | 0.33s (.626) | 0.50s (.52) | 5.56s (.712) | 0.21s (.452) | 24.28s (.826) | 302.94s (.916) |

Table 3: Inference times on test set. The size column shows the possible prune states representing the number of nodes in each layer of the neural network, each state 50% smaller than the last. Then for each dataset we provide the inference time on the test set in seconds followed by the percentage improvement from [1600, 800] in parentheses.

| Size | Alcohol | Games | Wine | Chocol. | Poker | Titanic | Health | Susy |
|------|---------|-------|------|---------|-------|---------|--------|------|
| **1600, 800** | 0.06s (.00) | 0.82s (.00) | 0.08s (.00) | 0.11s (.00) | 213.77s (.00) | 0.06s (.00) | 4.22s (.00) | 33.83s (.00) |
| **800, 400** | 0.06s (.04) | 0.7s (.145) | 0.08s (.087) | 0.1s (.077) | 220.94s (-.034) | 0.06s (.048) | 3.79s (.101) | 24.12s (.287) |
| **400, 200** | 0.06s (.038) | 0.7s (.153) | 0.07s (.101) | 0.1s (.09) | 188.36s (.119) | 0.05s (.17) | 3.71s (.12) | 21.54s (.363) |
| **200, 100** | 0.05s (.105) | 0.68s (.175) | 0.09s (-.083) | 0.1s (.093) | 166.78s (.22) | 0.05s (.159) | 4.03s (.044) | 22.75s (.328) |
| **100, 50** | 0.05s (.11) | 0.68s (.169) | 0.07s (.118) | 0.1s (.098) | 154.82s (.276) | 0.05s (.177) | 4.04s (.041) | 21.94s (.352) |
| **50, 25** | 0.05s (.11) | 0.7s (.143) | 0.07s (.136) | 0.1s (.109) | 153.34s (.283) | 0.05s (.169) | 4.03s (.045) | 20.55s (.393) |
| **25, 13** | 0.05s (.111) | 0.7s (.146) | 0.08s (.1) | 0.1s (.124) | 132.77s (.379) | 0.06s (.139) | 4.09s (.03) | 19.3s (.43) |
| **13, 7** | 0.05s (.118) | 0.68s (.169) | 0.07s (.143) | 0.1s (.11) | 105.15s (.508) | 0.06s (.109) | 4.06s (.037) | 19.7s (.418) |
| **7, 3** | 0.05s (.122) | 0.68s (.174) | 0.07s (.123) | 0.1s (.12) | 105.67s (.506) | 0.05s (.184) | 4.03s (.044) | 20.57s (.392) |
| **3, 1** | 0.06s (.044) | 0.68s (.175) | 0.08s (.085) | 0.1s (.1) | 106.74s (.501) | 0.05s (.184) | 3.65s (.135) | 22.04s (.349) |
| **1, 1** | 0.05s (.083) | 0.68s (.175) | 0.07s (.156) | 0.1s (.145) | 103.05s (.518) | 0.05s (.156) | 3.59s (.148) | 20.54s (.393) |

Table 2 and Table 3 show the time taken to run the tabular models. The first table contains the training times averaged over all epochs while the second contains the inference time on the test set. The tables present the number of seconds taken and the reduction of time in percentage compared to the largest model. The trend shows that a smaller model can lead to faster training and faster inference until we reach extremely small models of a few nodes which show some noise of a few milliseconds. The small datasets of a few hundred samples don't provide much insight, but the larger datasets show a better downtrend. The computations were made on a 2 CPU systems for all tests, no use of a GPU.

Table 4 highlights our best performing models accuracy-wise. In bold are the models with the best accuracy. The table highlights the RMSE or F1, the size of the model, the difference in percentage compared to the original model, and the pruning mode that generated the result. Iterative on the largest model (approach 1) has the best likelihood of generating a better model with 5/8 datasets improving. Iterative on the best performing model (approach 2) has one best case tied with approach 1 for the poker dataset, but it happens that the largest model was the best performing model, so both approaches generated the same result. Oneshot had one case of generating a best performing model for the smallest dataset in our experiments. There were two datasets, Wine and Chocolate, which could not be pruned further without some loss in accuracy, but they can be pruned extremely small with some degradation in accuracy.

We compare the tabular models to other models in Table 5 and found that even the original model has the best performing accuracy for all but one result. In this case the Health dataset using SVR resulted in a better accuracy than the original model, but not better than our pruned models. All comparisons shown are to the original tabular model. The table shows the following models: K Nearest Neighbors (KNN), Linear Regression (LR), Epsilon-Support Vector Regression (SVR), Gradient Boosting

Table 4: Comparison of accuracy (RMSE/F1) for each dataset between the original tabular models, and iterative/oneshot modes. First result is the best performing original models, second result is best performing iterative when pruning the best original model, third result is best performing iterative when pruning the largest model [1600, 800], and the final result is the best performing oneshot model. Results are shown in RMSE/F1 depending on the dataset along with the model size, then we include the difference in percentage along with the pruning mode for the model. Bold is marked as best performing accuracy.

| Dataset | Best Original | Iterative (from best original) | | Iterative (from [1600, 800]) | | Oneshot | |
|---|---|---|---|---|---|---|---|
| | Acc Size | Acc Size | Diff% Mode | Acc Size | Diff% Mode | Acc Size | Diff% Mode |
| **Alcohol** | 0.903567 3 | 0.895091 1 | +0.937% L1 | 0.899411 1 | +0.459% LN | **0.89357 1, 1** | **+1.106% LN/Rand** |
| **Games** | 0.3039825 13 | 0.3013613 7 | +0.862% L1 | **0.26265 100, 50** | **+13.596% LN** | 0.45927 1, 1 | -51.086% Rand |
| **Wine** | **0.59206 100, 50** | 0.5993513 7 | -1.231% L1 | 0.5986325 13 | -1.11% LN | 0.61947 1, 1 | -4.63% Rand |
| **Chocol.** | **0.46411 200,100** | 0.483311 1 | -4.137% All | 0.48291 200,100 | -4.051% LN | 0.48117 1, 1 | -3.676% L1/LN |
| **Poker** | 0.56241 1600,800 | **0.53551 400,200** | **+4.783% L1** | **0.53551 400,200** | **+4.783% L1** | 0.76128 1, 1 | -35.36% L1 |
| **Titanic** | 0.78571 13,7 | 0.785717 3 | +0.000% LN | **0.7931 13, 7** | **+0.941% L1** | 0.78333 1, 1 | -0.303% L1/LN |
| **Health** | 0.82263 800,400 | 0.823227 3 | +0.072% LN | **0.82428 200,100** | **+0.201% Rand** | 0.82242 1, 1 | -0.026% LN |
| **Susy** | 0.77080 200,100 | 0.77071 100,50 | -0.012% Rand | **0.77109 200,100** | **+0.038% L1** | 0.75316 1, 1 | -2.289% L1/LN |

Regressor (GBR), Decision Tree (DT) and Random Forest (RF). We also used the equivalent classifier version of these models for the classification tasks. All models were tested on the exact same train, validation and test sets and we also preprocessed the data using in the same way converting text/categorical features to values, normalizing continuous features and filling in missing values. KNN was optimized for K using the validation set, GBR and RF used 100 estimators and SVR used the RBF kernel as defaults selected by scikit-learn. Note that Susy has KNN and SVR missing, this is because even with a 32 CPU machine and weeks of computation we couldn't produce a result in time, so we omitted them from the table.

In comparison with other approaches, some datasets do not have reported results such as Video Games, Alcohol and Chocolate. The Titanic dataset is shown on kaggle to achieve 100% on the test set using random forest and other approaches due to an extended features dataset derived from the literature. Without using extended features, running the same model as in our best result (L1 iterative [13, 7]) on kaggle's test set gives us a score of 0.77511. Because of missing available comparisons, we provide comparisons using exactly the same data split and preprocessing for all datasets using several models in Table 5 as mentioned previously.

For the Red Wine dataset, Dahal et al. (2021) reports the following RMSE results: 0.63245 using a 3 layer neural network, 0.61163 using GBR, 0.62145 using SVM, and 0.62201 using ridge regression. In this case we hold the best result with 0.59206 RMSE (+3.200%) using an original tabular model sized at [100, 50].

Şekeroğlu (2021) reports several F1 scores on the Health dataset: 0.80 using KNN, 0.77 using naïve bayes, 0.76 using LR, 0.81 using RF, 0.80 using MLP, 0.72 using SVM. Our best model using

Table 5: Comparison of different models to the original tabular model in Table 4, top is RMSE/F1 depending on the dataset and bottom is the difference in percentage compared to the original tabular model. In all but one case (Health SVR) the original tabular model outperforms all of these models. Note that the pruned tabular models are not reflected in these results and the difference is only computed for original.

| Dataset | KNN | LR | SVR | GBR | DT | RF |
|---|---|---|---|---|---|---|
| **Alcohol** | 0.94405 | 0.96838 | 0.94680 | 0.92735 | 1.08500 | 0.94940 |
| | -4.481% | -7.174% | -4.786% | -2.633% | -20.081% | -5.073% |
| **Games** | 1.0253 | 0.61135 | 0.98243 | 0.69862 | 0.70800 | 0.68026 |
| | -237.292% | -101.115% | -223.189% | -129.824% | -132.91% | -123.784% |
| **Wine** | 0.65610 | 0.62515 | 0.59983 | 0.62044 | 0.81586 | 0.60776 |
| | -10.816% | -5.589% | -1.312% | -4.793% | -37.800% | -2.652% |
| **Chocolate** | 0.51295 | 0.51541 | 0.50163 | 0.50100 | 0.63895 | 0.48868 |
| | -10.523% | -11.053% | -8.084% | -7.949% | -37.672% | -5.294% |
| **Poker** | 0.73195 | 0.77344 | 0.74685 | 0.71312 | 1.06652 | 0.67302 |
| | -30.145% | -37.522% | -32.795% | -26.797% | -89.634% | -19.667% |
| **Titanic** | 0.68852 | 0.73504 | 0.50000 | 0.73214 | 0.68421 | 0.74380 |
| | -12.370% | -6.449% | -36.363% | -6.818% | -12.918% | -5.334% |
| **Health** | 0.79414 | 0.82106 | 0.82367 | 0.82137 | 0.71185 | 0.80276 |
| | -3.463% | -0.191% | +0.126% | -0.153% | -13.467% | -2.415% |
| **Susy** | | 0.68483 | | 0.76511 | 0.69232 | 0.76648 |
| | | -11.153% | | -0.738% | -10.182% | -0.560% |

random iterative pruning (approach 1) and a size of [200, 100] achieves an F1 score of 0.82428 (+1.763%).

Table 6: Best models selected by size with less than 2% divergence in accuracy of the original model. For each dataset (first column), we noted the size of the original model in the second column. Then we show the smallest possible model with <2% divergence in accuracy for each of the original, iterative on best original, iterative on [1600,800], and finally oneshot. The table shows the percentage difference in RMSE or F1, and P the prune rate compared to original. If the difference in accuracy is >2%, then that was the best performing model accuracy-wise and we could not produce a valid smaller model using that approach. Bold is the smallest model which doesn't exceed the 2% divergence in accuracy rule.

| Dataset | Orig. | Original <2% | Iterative <2% (best orig.) | Iterative <2% ([1600,800]) | Oneshot |
|---|---|---|---|---|---|
| | **Size** | **Diff (P)** | **Diff (P)** | **Diff (P)** | **Diff (P)** |
| **Alcohol** | 7, 3 | -0.923% (.600) | +0.937% (.800) | +0.459% (.800) | **+1.106% (.800)** |
| **Games** | 25, 13 | **-0.158% (.737)** | +0.862% (.474) | +4.168% (-.974) | -51.086% (.947) |
| **Wine** | 100, 50 | 0.000% (.000) | **-1.231% (.867)** | -1.110% (.747) | -4.630% (.987) |
| **Chocol.** | 200,100 | **0.000% (.000)** | -4.137% (.993) | -4.441% (.993) | -3.676% (.993) |
| **Poker** | 1600,800 | -0.768% (.969) | **-0.349% (.984)** | **-0.349% (.984)** | -35.360% (.999) |
| **Titanic** | 13, 7 | 0.000% (.000) | -1.754% (.800) | **-0.303% (.900)** | **-0.303% (.900)** |
| **Health** | 800,400 | -0.085% (.997) | -0.188% (.998) | -0.287% (.998) | **-0.026%(.998)** |
| **Susy** | 200,100 | -1.758% (.987) | **-0.381% (.987)** | -0.397% (.987) | -2.289% (.993) |

Museba et al. (2021) reports accuracy as their metric and achieve the following results on the Poker dataset: 0.7687 using the Heterogeneous Dynamic Ensemble Selection based on Accuracy and Diversity (HDES-AD) approach, and 0.9068 using their HDES-ADP variant. They also report a comparison to Diversity for Dealing with Drifts (DDD) Minku & Yao (2011) with 0.7867, Online

Accuracy Updated Ensemble (OAUE) Brzezinski & Stefanowski (2014) with 0.7325, and Active Fuzzy Weighting Ensemble (AFWE) Dong et al. (2018) with 0.7126. These approaches all use techniques to train ensembles of models and in particular Museba et al. (2021) generates models based on diversity and accuracy. We used RMSE instead of accuracy, but reran our models using F1 to optimize the models for this comparison. We achieved 0.87915 F1, 0.88360 accuracy (-2.558%) using L1 iterative [800, 400] (approach 1), and a similar score with a smaller model 0.87958 F1, 0.88462 accuracy (-2.446%) using Random iterative [400, 200] (approach 1).

Azhari et al. (2021) uses a 70/30 train-test split on the Susy dataset at random. We used the recommended test set provided for our results of 10% in size using the final 500k samples in the list and a validation set of 10%. They report accuracy as their metric with the following scores: 0.7884 using LR, 0.774 using RF, 0.7546 using DT, 0.793 using Gradient Boosted Tree (GBT). We achieved a better result of 0.77109 F1 using L1 iterative [200, 100] (approach 1), and an accuracy of 0.80285 (+1.242%).

Finally as our last result, we show a comparison of model size to the original tabular model by selecting the smallest model we can with less than 2% divergence in RMSE or F1. We do the same with the original models, so we select the best size for original with less than 2% divergence in accuracy from the best performing RMSE/F1 original model. Then we apply the same rule to iterative and finally oneshot. The results are shown in Table 6 highlighting in bold the smallest model following the described conditions. We show the original model size, then we present the difference in accuracy of other models and the prune rate from the same original model. In 7 out of 8 datasets we can generate a smaller model with minimal affect to RMSE/F1 including original with selection. In 6 of 8 datasets we outperform original with selection for 2% divergence generating models over 85% smaller and many over 98% smaller.

# 6 CONCLUSION AND FUTURE WORK

In conclusion, we presented two approaches to pruning tabular neural networks, iterative and oneshot, based on the lottery ticket hypothesis using structured node pruning. The results are presented for 8 tabular datasets of different sizes and feature sets. We improved accuracy in 6 of the 8 datasets when considering accuracy alone. We also show up to 85% reduction in nodes in 6 of 8 datasets considering model size with limited affect to RMSE/F1 and over 98% reduction for many of them. We show that the tabular models outperform several other tabular models where comparisons weren't available such as KNN, RF, SVR, DT, LR and GBR. Finally, when comparing to other papers, we show an improvement of +3.200% in RMSE for the Wine dataset, +1.763% in F1 for the Health dataset and +1.242% in accuracy for the largest dataset of 5 million samples, Susy.

We found that the iterative approach while pruning a large model obtains the majority of top results compared to pruning better but smaller models, or in comparison to oneshot pruning. We go beyond just masking weights and implement a structured pruning approach reducing the model architecture to layer sizes as low as [1, 1] while improving accuracy in comparison to large model sizes like [1600, 800] for the alcohol dataset. Finally, for each dataset, we show the advantage in training and inference time of structured pruning at each pruning state. Future work will be focused on pruning layers once the size of a layer reaches one node, and other focuses will be on feature selection using the lottery ticket hypothesis where pruning nodes responsible for certain features will result in the removal of that feature.

We asked the question "if these smaller networks perform just as well as the larger ones, could one not train the smaller network architecture to begin with?", and we believe it is clear that given the right set of initial weights for the network, a smaller, faster and better network can be trained to begin with. We show the existence of these weights through iterations of pruning, so perhaps there exists an approach to initializing the network as a function of the data using lottery weights.

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
