# OpenReview forum: "Lottery Ticket Structured Node Pruning for Tabular Datasets"
_ICLR.cc/2022/Conference — ICLR 2022 Submitted_

### Official Review · Reviewer_k3Jq · 2021-10-29

**Correctness:** 2
**Technical Novelty And Significance:** 2
**Empirical Novelty And Significance:** 1
**Recommendation:** 3
**Confidence:** 3

**Main Review:**

In terms of significance the problem-to-solve, I think compressing a tabular neural network is of at least modest importance. This is mainly because most existing works on pruning algorithms do not even consider networks designed for tabular datasets. In a sense, this is not too surprising---neural networks are usually not state-of-the-art methods on such datasets, the benefits of overparameterization are obscure in such domains, and the need for processing tabular data on edge devices is worse-motivated than vision/language/speech domains. Still, having a work that formally studies pruning models for tabular datasets will be a nice starting point for future works in the direction. But at this stage, I do not think this paper is ready to be published, due to the following reasons:

One of the biggest limitations of this paper is that the proposed method is only evaluated on a specific tabular neural network (namely the one provided by FastAI). Up to my knowledge, there are many different types of neural network architectures in the literature designed to process tabular data, including TabNet, TabTransformer and SAINT. Yet, authors evaluate the proposed pruning scheme only on the tabular neural networks that is included in the FastAI framework (which I am not sure about what the framework is implementing). Thus I am so sure if the methods will also perform well on other model architectures, or these methods are very specialized to a specific model. If the latter is the case, I expect the impact of this work to be very small, and is more appropriate to be included as a manual section of the framework rather than an academic publication.

Another thing that worries me is the experimental validation of the method (and many design choices). Although the structured pruning pipeline with weight rewinding looked quite standard, there were some unusual choices, such as using the $L_{-\infty}$ criteria for pruning nodes. I am quite curious how much this choice contributes to the performance of the method (when compared to $L_1$, $L_2$, or $L_{\infty}$). If this is indeed a critical factor for the pruning performance, I think this novel finding could be of interest to a much wider audience. But as of now, I do not think the paper is well-validating the method or this specific component; the paper lacks comparisons to the baseline pruning methods, with/without rewinding and/or using different norms. Also, using $P=0.5$ looked quite extreme to me, as many previous iterative pruning works tend to prune much smaller fraction of weights in each iteration (e.g., the lottery ticket paper). I wonder if this has any critical impact on the final performance.

As a minor remark: The paper was not 100% clear to me, especially in section 4. Could the authors give a more formal statement of the algorithm (say, as a pseudo-code in "algorithm" environment)?

**Summary Of The Paper:**

This paper proposes two structured pruning strategies for compressing a (very specific) tabular neural network: iterative pruning, and one-shot pruning. Both methods work by removing the node with smallest $L_{-\infty}$ norm, i.e., the one with smallest minimum weight of each node. After removing a designated amount of nodes, the initial weights of the models are restored (as in Frankle et al. (2019)), and optionally proceeds for the next pruning iteration. The methods achieve noticeable reduction in the total training (wall-clock) time and inference time, while introducing only small degradation in performance.

**Summary Of The Review:**

The paper studies a fairly important problem, but I do not think that the proposed methods are validated well experimentally. Much more additional experiments will be needed to clarify the benefit of the proposed method.

---

### Official Review · Reviewer_Y7kX · 2021-10-31

**Correctness:** 4
**Technical Novelty And Significance:** 2
**Empirical Novelty And Significance:** 2
**Recommendation:** 5
**Confidence:** 4

**Main Review:**

The questions are outlined below:
- While I thank the authors for giving introductions to tabular datasets, the length of them seems too long - taking almost one page. Authors may consider putting some description into the supplementary materials.
- More experiment settings would be better. For example,  the authors use a very large pruning rate for one-shot pruning while for iterative pruning the rate is small. Maybe consider comparing one-shot pruning with P=0.5.
- The way where authors present their results seems a little confusing - especially putting RMSE and F1 together as accuracy. We prefer lower RMSE but higher F1, so the calculation of diff% is inconsistent in Table 4 for example.
- Would the authors provide a detailed explanation on why we need sparse models for tabular datasets? Increasing the performance a little bit seems weak for me - is there any challenge like over-parameterization? We need many rounds of IMP to reduce the model size from [1600,800] to [1,1] while the inference time seems to only be reduced by 90%. More importantly, it seems that we can pick a small network from the beginning (if I have understood correctly), which means we don't need pruning to get compressed networks. I find this is an essential difference between tabular and other (CV, NLP, speech) models, i.e., the latter one doesn't permit such a big modification to the models' structure otherwise the performance is greatly harmed. However, maybe it is not the case on tabular models.


**Summary Of The Paper:**

This paper investigates model pruning and the Lottery Ticket Hypothesis in the context of tabular datasets and model training. The authors apply a set of pruning techniques to the tabular neural networks from FastAI and examine whether the LTH still holds on the tabular datasets and their corresponding models. Different pruning methods are used, namely different norms of selecting the parameters to be pruned, to see if the winning tickets exist. The results suggest that on some datasets the winning ticket exist, and outperform classical methods like random forest.

**Summary Of The Review:**

The experiments and the methods are clear in this paper, occasionally with some problems which I have mentioned above. From my perspective of view, the main problem here is motivation. I would give a score of 5 currently, but I am also willing to change my score if the authors provide useful responses.

---

> ### Comment · Reviewer_Y7kX · 2021-11-29
> **Final score**
>
> Since there is no feedback from the authors, I decide to keep my score.

---

### Official Review · Reviewer_EuY8 · 2021-11-02

**Correctness:** 3
**Technical Novelty And Significance:** 2
**Empirical Novelty And Significance:** 2
**Recommendation:** 3
**Confidence:** 3

**Main Review:**

While I find interesting the topic addressed, below are some suggestions for improvement:
- In the Introduction section try to present in a concise manner which are the paper novel contributions in comparison with state-of-the-art
- Improve Related Work on sparse neural networks for tabular data. There are some papers addressing this topic. You can start by reading some recent survey papers (e.g., [1]) and some concrete approaches which use sparse neural networks for tabular data (e.g., [2,3]).
- From Table 1, I can see that the datasets studied have quite a low number of features. To properly validate the proposed method, I suggest also trying datasets with a much larger number of features.
- Add some algorithms to describe the proposed methods in the Methodology section. Also, please add equations with the pruning criteria.
- Provide the code developed as open-source for easy reproducibility.
- Please improve the readability of the Results section, while trying to have a more structured presentation and better qualitative discussions. Currently, it is hard to follow all the details and to understand which is the actual improvement of the proposed method with respect to the state-of-the-art. Preparing some summary plots for the whole experimental evaluation may help. Also, please extend the baseline methods used for comparison to cover well the state-of-the-art.
- The current writing style of the paper does not follow the academic writing style and it is more like a report. Please try to improve it.
- There have been previous attempts to address the question from the last paragraph of the Conclusion section. Non-exhaustively, please see for instance [1,2,3]. I suggest trying to rephrase it to make it more accurate.

References:
- [1] Hoefler et al., Sparsity in Deep Learning: Pruning and growth for efficient inference and training in neural networks, JMLR 2021, https://www.jmlr.org/papers/v22/21-0366.html
- [2] Liu et al., Sparse evolutionary deep learning with over one million artificial neurons on commodity hardware, NCAA 2020, https://link.springer.com/article/10.1007/s00521-020-05136-7
- [3] Peterson et al., Cognitive model priors for predicting human decisions, ICML 2019, https://arxiv.org/abs/1905.09397


**Summary Of The Paper:**

This paper proposes two approaches to use sparse neural networks on tabular data. The topic is interesting and sometimes overlooked in the literature. The proposed approaches are based on node (likely referring here to neuron) pruning. Overall, I believe that the proposed method description, experimental evaluation, and paper writing quality can be improved.

**Summary Of The Review:**

Overall, even if it offers some interesting perspectives, I believe that this paper is immature and not ready yet for publication.

---

### Official Review · Reviewer_EcHe · 2021-11-02

**Correctness:** 3
**Technical Novelty And Significance:** 1
**Empirical Novelty And Significance:** 1
**Recommendation:** 1
**Confidence:** 5

**Main Review:**

Strengths:
- Comprehensive set of experiments on a tabular network and corresponding datasets using a single dataset

Weaknesses:
- Lack of novelty; see below
- Lack of clarity and experimental details.


**Summary Of The Paper:**

This paper presents an empirical study of pruning algorithms on tabular neural networks using the lottery ticket hypothesis. A comprehensive set of results show that iterative pruning seems to work better compared to one-shot pruning.

**Summary Of The Review:**

My major concern for this paper is the lack of novelty and lack of experimental details. In the end, this is an empirical work providing no novelty. Those details should be extremely clear.

Novelty:
- The introduction claims the paper proposes 2 pruning methods; However, in the method section, there are only references to two pruning strategies: iterative and one shot. These two strategies are well-known and utilized in the pruning literature; Iterative pruning was initially presented at Han et al. Deep Compression: Compressing Deep Neural Networks with Pruning, Trained Quantization, and Huffman Coding.
For one-shot, also referred as single-shot pruning, Lee et al. SNIP: SINGLE-SHOTNETWORKPRUNING BASED ONCONNECTIONSENSITIVITY, ICLR 2019
The paper does not refer to any approach in the literature. More importantly, pruning is not only about strategy but also the saliency metric used to remove parameters.




On the experimental side:

- For instance, the paper describes two modalities of pruning (iterative vs one-shot) but does not describe the type of metric used to measure the saliency of each neuron / parameter. Algorithmic details here would be beneficial. There is a reference to the type of norm but seems like those are not details. Would be also good to show numbers for importance based methods (see for instance Molchanov et al. Importance Estimation for Neural Network Pruning, CVPR2019)
- Network architectures: The algorithm uses the results from FastAi; What are the details of these architectures? Would be good to add some additional architectures to have more conclusive results.
- Tables seem to be too large (for instance the training time). Adding better tables / graphs would leave space to have these details in place.

- The paper focuses on the lottery hypothesis; However, the ultimate goal is to prune a network as much as possible while maintaining accuracy. Therefore, I would expect comparisons to other approaches not based on the lottery ticket: importance-based pruning; magnitude, prune while training, zero shot pruning.....



- How are models with less than 2% accuracy selected? I guess that is an empirical selection, right?


Clarity:
- Would be more readable if the paper explains the lottery hypothesis and related pruning works and how those approaches can not be applied to tabular data; In the current form, seems like it is just plug and play: The paper states: We focus on applying this hypothesis to tabular neural networks. Why is this challenging? Why is this different to applying to other data / networks? What has been done in that space in terms of pruning / compression?

---

> ### Comment · Reviewer_EcHe · 2021-12-02
> **Final rating**
>
> There is no rebuttal, so I keep my score

---

### Decision · Program_Chairs · 2022-01-20

**Decision:**

Reject

**Comment:**

### Summary

The paper demonstrates the applicability of pruning to tabular datasets, which aren't typically explored in the literature on pruning. The work identifies that yes, pruning can indeed be applied to this domain with some success.


### Discussion

#### Strengths

An unconventional domain that, nonetheless, should be studied.

#### Weaknesses

The empirical setup does not include comparisons to baselines or ablations (e.g., different importance metrics).

### Decision

I recommend Reject. Reviewer k3Jq provides a precise and constructive set of criticisms that if solved would make for an interesting and significant piece of work.